# Colon Fibroblasts and Inflammation: Sparring Partners in Colorectal Cancer Initiation?

**DOI:** 10.3390/cancers13081749

**Published:** 2021-04-07

**Authors:** Lauriane Onfroy-Roy, Dimitri Hamel, Laurent Malaquin, Audrey Ferrand

**Affiliations:** 1IRSD, Université de Toulouse, INSERM, INRA, ENVT, UPS, 31024 Toulouse, France; lauriane.roy@inserm.fr (L.O.-R.); dimitri.hamel@inserm.fr (D.H.); 2LAAS-CNRS, Université de Toulouse, CNRS, 31024 Toulouse, France; laurent.malaquin@laas.fr

**Keywords:** colon, fibroblasts, stroma, colorectal cancer, inflammation, intestinal stem cells

## Abstract

**Simple Summary:**

Colorectal cancer (CRC) is the third most common cause of cancer-related death. Patients suffering inflammatory bowel disease have an increased risk of CRC. It is admitted that CRC found its origin within crypts of the colon mucosa, which host the intestinal stem cells (ISCs) responsible of the tissue renewal. ISC behavior is controlled by the fibroblasts that surround the crypt. During inflammation, the signals delivered by fibroblasts are altered, leading to stem cells’ dysregulation, possibly turning them into cancer-initiating cells. Here, we reviewed the interplays between the fibroblast and the ISCs, possibly leading to the initiation of CRC due to chronic inflammation.

**Abstract:**

Colorectal cancer (CRC) is the third most common cause of cancer-related death. Significant improvements in CRC treatment have been made for the last 20 years, on one hand thanks to a better detection, allowing surgical resection of the incriminated area, and on the other hand, thanks to a better knowledge of CRC’s development allowing the improvement of drug strategies. Despite this crucial progress, CRC remains a public health issue. The current model for CRC initiation and progression is based on accumulation of sequential known genetic mutations in the colon epithelial cells’ genome leading to a loss of control over proliferation and survival. However, increasing evidence reveals that CRC initiation is more complex. Indeed, chronic inflammatory contexts, such as inflammatory bowel diseases, have been shown to increase the risk for CRC development in mice and humans. In this manuscript, we review whether colon fibroblasts can go from the main regulators of the ISC homeostasis, regulating not only the renewal process but also the epithelial cells’ differentiation occurring along the colon crypt, to the main player in the initiation of the colorectal cancer process due to chronic inflammation.

## 1. Introduction

Despite its reputation as a curable disease, and a thorough characterization of the mutations involved in its adenoma–carcinoma sequence [1,2,3], colorectal cancer (CRC) remains worldwide the third most common cause of cancer-related death [4]. To date, the debate about the origin of the CRC stem cells remains largely open. Two theories have been proposed. The bottom-up theory, where the intestinal stem cells (ISCs) are the cells of cancer origin, and the top-down theory, where the cancer initiates in progenitors or differentiated cells. However, while the latter originated mainly from histopathological observations, recent reports strengthened the bottom-up theory. ISC-specific deletion of both adenomatous polyposis coli (APC) alleles using either Bmi1-, CD133-, or LGR5-Cre recombinase mice leads to a rapid full adenoma formation [5,6,7], while a similar APC deletion in late progenitors or differentiated cells only results in sporadic and slow-developing adenoma [7]. Therefore, the scientific community agrees on the idea that the crypt is the place where the cancer originates.

Indeed, the fact that almost all epithelial cells in the intestinal lining are replaced on a weekly basis puts great demands on the cellular organization of this tissue and represents, in consequence, a high risk of malignant conversion. The renewal of the intestinal epithelium is maintained by an ISC compartment that resides at the bottom of the crypt (Figure 1). It depends on the spatial organization of signals emanating from the supportive mesenchymal cells, as well as from differentiated epithelial progeny. However, the high number of patients developing CRC indicates that these regulatory mechanisms often fall short in protecting against malignant transformation. Fearon and Vogelstein have clearly demonstrated that CRC develops as a stepwise accumulation of genetic hits in specific genes and pathways [1]. The cancer stem cell theory refines this model and suggests that the actual tumorigenic capacity of individual cancer cells may be influenced by homeostatic signals derived from their microenvironment [8].

In fact, CRC is a disease in which the homeostatic capacities of intestinal crypt stem cells are impaired. Indeed, under physiological conditions, the homeostasis and integrity of the crypt is tightly controlled by a very specific environment, namely the intestinal crypt niche [9]. This niche includes the basal lamina upon which reposes the crypt and the stroma surrounding the crypt and including the extracellular matrix and the stromal cells (fibroblasts; neurons; and glial, immune, and vascular cells). The intestinal stroma’s main functions toward the epithelium, beside immunity, are mechanical (support) and metabolic (nutrition and various exchanges). Among the stroma, the fibroblasts that sheathe the crypt are key regulators of the crypt cells by secreting factors such as wingless-related integration site (Wnt) ligands, R-spondin, Noggin, and bone morphogenetic protein (BMP) that control stem cell phenotype and differentiation processes along the crypt. Thus, in order to preserve a good control of the crypt homeostasis and integrity, and thus avoid any risk of tumoral transformation, the integrity of each partner, the niche as well as the crypt stem cells, has to be preserved. This implies that any alteration of one or another of the crypt or niche entities could in turn impact on the homeostasis of the colonic mucosa and, thus, favor cancer initiation [10].

To date, little is known on the mechanisms implicated in the stroma-dependent deregulation of the crypt stem cells. Compared to the healthy intestinal stroma, the inflammatory and the tumor stroma have distinct pathological features and are said to be “activated” with continuous remodeling. The predominant cell type in the stroma is the fibroblast. The functions of fibroblasts include the deposition of extracellular matrix (ECM), regulation of epithelial differentiation, regulation of inflammation, and involvement in wound healing (Figure 2). As the principal source of ECM constituents, fibroblasts are considered the main mediators of scar formation and tissue fibrosis. In an activated stroma state, activated fibroblasts present a different phenotype and secrete different types of chemokines, cytokines, proteases, and ECM proteins than in their normal non-activated conditions [11]. Activated fibroblasts are involved in the proliferation and dedifferentiation of tumor epithelial cells and can participate in tumor resistance for example by increasing the ECM stiffness around the tumor, or by upregulating the epithelial stemness capacities and favoring the epithelial–mesenchymal transition in vivo [11,12,13]. Thus, through the secretion of numerous factors, activated fibroblasts, including the cancer-associated fibroblasts (CAFs), contribute to the creation of an environment favoring the tumor development not only by regulating the reorganization of the connective tissue but also through tumor neo-angiogenesis allowing metastasis.

Nowadays colorectal CAF roles have started to become well documented. However, despite evidences reporting the role in cancer initiation of chronic alteration of colon-resident fibroblasts due to a long-term or chronic inflammatory context, the cellular and molecular mechanisms involved remain less understood. In this manuscript, we decided to review the interaction between the fibroblast and the colorectal epithelial cells not only during homeostasis but also during fibrosis and inflammation with the aim of discussing how fibroblast alteration can foster ISC dysregulation and favor colorectal cancer initiation.

## 2. From Normal to Activated Fibroblasts

### 2.1. Fibroblasts Act as ISCs’ Nanny

From an overall perspective, fibroblasts within the colon stroma are either sparsely disseminated within the ECM or sheathing the crypt (Figure 3). Increasing numbers of studies portray fibroblasts as crucial players in the ISC niche. These subepithelial fibroblasts are very heterogeneous. Along the crypt, we could distinguish at least two main peri-epithelial fibroblast populations. The subepithelial fibroblasts located around the bottom of the crypt are an important source of Wnt ligands, which regulate ISC renewal, and BMP antagonists, blocking the differentiation process. Fibroblasts located at the top of the crypt are associated with the epithelial differentiation process by inducing BMP pathway activation [14,15]. Using in situ hybridization and RT-PCR on human colon tissue, Kosinski et al. observed that BMP antagonists such as Noggin, Gremlin 1, Gremlin 2, and Chordin-like 1 are expressed by myofibroblasts expressing vimentin (VIM) and alpha smooth muscle actin (αSMA) and located at proximity of the bottom of the crypt [16]. αSMA^+^ cells also expressed Wnt2b and Wnt5a [17]. However, using single-molecule-RNA fluorescence in situ hybridization (FISH), authors demonstrated that αSMA^−^ cells strongly expressed those isoforms. These last years, numerous studies, performed on either the mouse or human small intestine or colon, aimed to decipher pericryptic fibroblasts’ heterogeneity by identifying many other non-myofibroblast markers present in the ISC niche. These include Foxl1 [18,19], Gli1 [20], platelet-derived growth factor receptor (PDGFR) α [21,22], CD90 [23], or gp38 [24,25]. Precisely, Stzepourginski et al. observed that a population of mesenchymal cells identified as gp38^+^ (first fibroblast marker described in literature also known as podoplanin, (PDPN)), in combination with CD34^+^ but αSMA^−^, are the main producers of Wnt2b, as well as R-spondin 1, and Gremlin 1 (GREM1), an inhibitor of the pro-differentiative BMP pathway [25]. In addition to αSMA and vimentin, PDGFRα has been proposed as a marker of subepithelial pericryptic fibroblasts in mice and humans [26]. In fact, combinations of cells positive for PDGFRα, gp38, Gli1, and CD90 cells were shown to support mice intestinal organoid growth in co-culture in contrast to the PDGFRα^+^, gp38^+^, Gli1^+^ but CD90^-^ population [23]. Using genetic ablation of porcupine acyltransferase (porcn) in PDGFRα^+^ pericryptal stromal cells, resulting in the loss of global epithelial Wnt secretion, Greicius et al. proposed those cells as the major in vivo source of Wnts in the murine intestine [21]. They also demonstrate that PDGFRα^+^ cells provide R-spondin 3, an important co-activator of Wnt/β-catenin signaling in ISC. Recently, using single-cell RNA-seq and whole-mount high-resolution microscopy, McCarthy et al. identified three distinct pericryptic mesenchymal cell types in the mouse small intestine based on the level of expression of PDGFRα. They identify a low PDGFRα subpopulation (called trophocytes) expressing CD81 at the crypt bottom that supports ISC proliferation in organoid co-culture assay. RNA-seq data reveal that contrary to villus-associated telocytes highly expressing PDGFRα, trophocytes suppress BMP signaling via high GREM1 expression levels while promoting ISC proliferation by expressing Wnt ligands (Wnt2b and 9a) and *RSPOND 1, 2, and 3* genes [22]. These data confirmed by Brugger et al. [27], taken as a whole, show that the characterization of the fibroblast populations involved in the ISC niche still remains largely incomplete, with some publications even reporting contradictory findings.

Interestingly, in addition to soluble factors secreted by the fibroblasts, Oszvald et al. showed on primary human and mouse organoids that extracellular vesicles derived from normal primary fibroblasts contain Wnt family factors and amphiregulin (AREG), allowing an increased ISC proliferation and enrichment by activating Wnt and Epidermal Growth Factor (EGF) pathways. Unfortunately, the fibroblasts producing these vesicles remain unidentified, with no markers specifying them [28]. To go further regarding fibroblast heterogeneity, by performing single-cells analysis in human and mice colons, Kinchen et al. identified four fibroblast subpopulations [29]. They found two populations, S1 and S2, present under the physiological condition and characterized by an enrichment in genes linked to ECM constituents. While S1 fibroblasts express non-fibrillar collagens (COL14A1, COL15A) and elastic fibers (FBLN1, FBLN2, FBLN5, EFEMP1, FN1), S2 fibroblast express collagen IV isoforms (COL4A5, COL4A6), suggesting a predominant role in the building of the interstitial matrix versus the basement membrane, respectively. S2 is also characterized by a high expression of transforming growth factor (TGF)-β superfamily ligands, BMP2 and BMP5, and non-canonical Wnt ligands, WNT5A and WNT5B; therefore, participating in epithelial homeostasis. In accordance with this hypothesis, S1 fibroblasts are localized throughout the lamina propria, whereas S2 mesenchymal cells are located in close proximity to the epithelial lining [29]. The S2 population can actually be divided into two subtypes. The S2a subpopulation is associated with the differentiation process, driving BMP signaling and response, while the S2b pericryptic subpopulation regulates ISC proliferation and niche establishment; it is also involved in wound healing. More specifically the CD142^+^/F3^+^ S2 population co-cultivated with human normal colonic organoids induces an ISC enrichment and non-budding (i.e., immature) organoid structures.

The S3 and S4 populations are almost absent from the healthy colon and mainly associated with disease states such as ulcerative colitis (UC), while the S2 population is lost in UC patients [29]. The S4 population, characterized by the combined expressions of CD24, CD74, and PDPN, emerges in UC patients and is associated with cytokine-mediated signaling pathways. Indeed, this S4 population expresses a high level of interleukin (IL)-33 and T cell co-stimulatory TNF-superfamily ligand (TNFSF14/LIGHT), these factors being proposed by the authors to promote the expansion of active LGR5^+^ ISC and reserve stem cells (expressing SOX9 and MSI1) according to the results obtained on human colon organoid cultures. Finally, the S3 population, characterized by the expression of cyclooxygenases (COX) 2 and the transcription factor B-cell lymphoma (BCL) 6, is associated with the organization of the ECM.

### 2.2. Fibroblasts Activation: When the Nanny Turns into the Nurse

Because of their major participation in the integrity of the ISC niche, fibroblasts have a leading role in intestinal epithelial lining repair in case of an injury and/or in an inflammatory context. Fibroblasts implicated in the establishment of a normal ISC niche play a concomitant role under the chronic inflammatory context to regenerate intestinal epithelium. Degirmenci et al. found that the Gli1^+^ stromal population (i.e., identified by authors to contribute to ISC niche) is increased in mice after dextran sulfate sodium (DSS) treatment compared to control. Gli1^+^ population enrichment improves RSPOND3 secretion and promotes ISC proliferation for epithelium healing [20]. Moreover, Harnack et al. demonstrated that, in addition to the physiological maintenance of the LGR5^+^ ISC in physiological conditions, under stress conditions, RSPOND3 acts via the LGR4 receptor expressed by progenitor and some specialized colonic cells to reactivate the expression of Wnt target genes to allow the regenerative process [30]. Moreover, to ensure wound healing, fibroblasts acquire an activated phenotype induced by various stimuli such as TGFβ [31,32] and PDGF [33], arising when tissue damage occurs and mainly released by epithelial cells or infiltrated monocytes and macrophages.

Activated fibroblasts are present under physiological conditions in normal colonic mucosa [34]; however, the population enrichment occurring during inflammation or tissue injury raises the question of the cellular origin of these newly activated fibroblasts. While it is well established that the activation of the resident fibroblasts is a great purveyor of activated fibroblasts, it has been shown that bone marrow stem cells can be another source [35,36,37]. Indeed, bone-marrow-derived fibroblasts from male mice can be detected via the detection of the presence of Y chromosomes, in the entire small intestine and colon subepithelial mucosa of the female recipients 7 days post-engraftment [35]. After trinitrobenzene sulfonic acid (TNBS)-induced colitis, bone-marrow-derived activated fibroblasts were significantly increased in the inflamed areas [36]. Moreover, this treatment can also induce an epithelial-to-mesenchymal transition (EMT), showing that the intestinal epithelial cells can provide new fibroblasts contributing to the fibrosis [38].

Compared to quiescent resident fibroblasts, activated fibroblasts display an altered protein expression profile allowing them to intervene in immune response, ECM remodeling, and epithelium regeneration (Figure 2). For example, normal adult human colonic subepithelial myofibroblasts, a subtype of activated fibroblasts expressing αSMA, express COX 1 and 2 and prostaglandin E2 (PGE2), known to be involved in the inflammatory response [39]. An increased COX2 expression is observed in response to IL-1α stimulation in human intestinal myofibroblasts and is frequently observed during acute and chronic intestinal inflammation [40]. This increase is mediated by the activation of multiple parallel signaling pathways implicating nuclear factor-κB (NF-κB), mitogen-activated protein kinases (MAPKs), extracellular signal-regulated protein kinase-1 or -2 (ERK-1/2), p38, c-Jun NH2-terminal kinase (JNK), stress-activated protein kinase (SAPK), and protein kinase C (PKC) [40]. The COX2–PGE2 pathway has a critical role in epithelium regeneration following DSS-induced colitis in mice via the activation of tumor progression locus-2 (Tpl2) kinase [41], a pro-inflammatory mediator, whose downregulation is genetically linked to inflammatory bowel diseases (IBD) [42]. Indeed, Tpl2 ablation in intestinal myofibroblasts results in enhanced ulceration and impaired compensatory crypt cells proliferation. Moreover, following epithelial injury and ISC loss, the stromal source of PGE2 has been proposed to play a crucial role in the recruitment of the epithelial reserve stem cells via the PTGER4 receptor combined with the activation of the Yes-associated protein (YAP) pathway to replenish the LGR5 active stem cell and, thus, improve the regeneration process [43]. These data show that under a regenerative context, the fibroblast-secreted PGE2 (in addition to RSPOND3) induces two epithelial responses to fully regenerate the intestinal lining: a dedifferentiation process and reserve stem cell recruitment, both allowing repopulation of the LGR5^+^ ISC compartment.

Moreover, activated fibroblasts also secrete fibroblast growth factor (FGF) described to regulate migration, proliferation, and renewal of the intestinal epithelial cells [44]. For example, in mice subjected to radiation injury, FGF2 is overexpressed by mesenchymal cells surrounding the ISC at the crypt bottom, and recombinant FGF2 injected before the irradiation increases the crypt stem cells’ survival [45]. FGF, established as a pro-angiogenic factor, must be secreted to avoid hypoxic issue in the regenerated tissue [46,47].

Another way used by activated fibroblasts to promote epithelium regeneration after an injury involves micro-RNA. Mi-RNA143 and mi-RNA145 have been demonstrated to repress insulin-like growth factor binding protein (IGFBP) 5, a negative regulator of insulin-like growth factor (IGF) signaling, in mesenchymal cells [48]. A targeted deletion of these mi-RNAs induce a failure in intestinal regeneration in vivo, highlighting their protective role.

Activated fibroblasts are crucial players in ECM remodeling as illustrated by their increased secretion of collagen in response not only to TNFα and IGF1 [49] but also matrix degradation through the production of matrix metalloproteinase (MMP). In response to respective IL-17 and IL-21 stimulation, human colonic myofibroblasts induce the expression and secretion of MMP3 [50] and promote the secretion of MMP1, 2, 3, and 9 in a dose-dependent manner [51]. This effect is cumulative with the TNFα-positive effect on those MMP secretions.

Thus, pericryptic fibroblasts answer epithelial alterations by rapidly adopting a specific secretory state in order to support dynamic processes necessary for ECM remodeling, immune cell activation, and epithelial regeneration. However, fibroblasts can be seen as the two sides of the same coin, initiating and sustaining wound healing on one side, while controlling and ending the tissue repair on the other side (Figure 3).

## 3. When the Good Guys Turn into Bad Guys

### 3.1. Wound Ending: How to Avoid the Drift?

Activated fibroblasts are key regulators of the immune response. They secrete a plethora of cytokines and pro-inflammatory molecules involved in the recruitment and activation of monocytes and macrophages. Nevertheless, they also have a role in the ending of the inflammatory response through the secretion of anti-inflammatory molecules [52]. With those mediators, activated fibroblasts can influence macrophages’ polarization toward M1/pro-inflammatory or M2/anti-inflammatory phenotypes [53]. Immunosuppression further relies on negative feedbacks utilizing the production of soluble mediators, such as inducible nitric oxide synthase (iNOS), by mesenchymal cells in response to a cocktail of pro-inflammatory cytokines IFNγ, TNFα, IL-1α, or IL-1β as demonstrated in mice [54].

Ending of wound healing is also mediated by apoptosis and deactivation of the activated fibroblasts by epigenetic changes [55]. While those regulatory mechanisms are extensively studied in organs such as skin, liver, or lung, little is known considering the gut. Nevertheless, one can suppose that the mechanisms involved remain quite similar [56]. For example, the combination of cytokines, such as IFNγ, TNFα, and IL-1β, induces a caspase-dependent apoptosis of human intestinal myofibroblasts [57]. IL-1β, or factors released during the inflammatory process such as HGF or bFGF, can trigger apoptosis of myofibroblasts in lung fibroses [58] and skin wounds [59], respectively. It has also been proposed that apoptosis of lung fibroblasts occurs after the withdrawal of the pro-survival growth factors TGFβ and PDGF during the later stage of wound healing [60,61]. The shift from tissue repair to pathogenic inflammation through myofibroblast apoptosis evasion has been recently reviewed [62]. To summarize, it seems that myofibroblasts are poised to self-destruct when tissue repair ends because of the absence of pro-survival signals. Nevertheless, some biomechanical and biochemical feedback loops can maintain those positive feeds; thereby, tipping the balance toward persistence. Another mechanism of cell death is speculated and relies on programmed necrosis through high production of COX2 when human dermal fibroblasts are clustered and cultivated in three-dimension spheroids [63]. However, another team did not observe such processes, thereby, keeping the question open [64].

Apoptotic mechanisms seem to not be exclusive, as cues arguing for deactivation and dedifferentiation processes of myofibroblasts are proposed, even if they remain largely unclear. bFGF induces the downregulation of αSMA through ERK1/2 regulation in skin fibroblasts [59]. Godichaud et al. proposed that trans-resveratrol is able to deactivate human liver myofibroblasts in vitro by reducing proliferation and migration and αSMA, MMP2, and collagen type 1 expression [65]. Hecker et al. found that the dedifferentiation of lung myofibroblasts is mediated through the ERK1/2–MAPK pathway promoting the downregulation of the myogenic transcription factor MyoD, implicated in their activation after TGFβ stimulation [66]. Furthermore, using a 3D scaffold built from collagen 1 and fibronectin, Sapudom et al. observed that human dermal fibroblasts activate in response to two days of TGFβ1 treatment but lose αSMA staining, decrease collagen 1 and fibronectin gene expression, and enhance cell motility after two days of IL-10 stimulation following TGFβ stimulation [67]. Interestingly, the loss of function of this cytokine induced IBD development in children [68].

A number of regulatory pathways exist in order to keep control on activated fibroblasts. Indeed, because of the multiple complex mechanisms involved from the inflammatory initiation to tissue repair, it is extremely important to tightly monitor in space and time each of the implicated actors to avoid serious issues.

### 3.2. Wounds That Do Not Heal: Persistent Fibroblast Activation in Chronic Inflammation

The inflammatory bowel diseases, two main subtypes of which are Crohn’s disease (CD) and ulcerative colitis (UC), are chronic, relapsing inflammatory disorders of the gastrointestinal tract. IBD afflicts several millions of persons worldwide. Their incidence is high among developed countries and increases steadily. These diseases are characterized not only by an excessive immune response to gut microbiota dysbiosis but also by an important impairment of the epithelial renewal process, leading to an abnormal mucosal repair/healing. Indeed, a major complication of UC is its evolution into cancer.

An issue in chronic inflammation is the persistence of an activated stroma. The phenotypic stability of activated fibroblasts in IBD and CRC is supported by the establishment of an autocrine signaling favorable to the maintenance of their phenotype. Leukemia inhibitory factor (LIF)-induced constitutive activation of the JAK1/STAT3 signaling pathway has been suggested to induce epigenetic changes both via DNMT3b DNA methyltransferase activation and continuous pro-inflammatory cytokine secretion [69]. Indeed, during chronic inflammation, IL-6, TNFα, and IL-1β maintain fibroblasts in an activated state [70,71]. Although transient fibroblast activation is mandatory to support the intestinal epithelium’s regeneration, a persistent activation promotes fibrosis, contributes to the persistence of the inflammation, as found in IBD, and later on, favors cancer initiation and progression. Indeed, cytokine signaling and immune response alteration are known to be associated with IBD and CRC [69,72,73].

However, the study by Kinchen et al. demonstrated that more than the disequilibrium between pro- and anti-inflammatory signals during chronic inflammation, it is the occurrence of an alteration in the fibroblastic subpopulation ratio that is linked to the disease state [29]. Indeed, the S4 fibroblast population, nearly inexistent in healthy patients, largely composed the fibroblast compartment in IBD patients at the expense of the S1 and S2 homeostasis-associated fibroblasts. As mentioned above, the S4 subpopulation expresses pro-inflammatory genes, such as lymphocyte trafficking cytokines (CCL19 and CCL21), TNFSF14/LIGHT, or IL-33 as well as markers such as PDPN. Interestingly, the expression levels of the pro-inflammatory cytokine oncostatin M (OSM) by macrophages and its receptor OSMR by PDPN^+^ myofibroblasts, are highly correlated with the degree of severity of IBD and is linked to anti-TNFα therapy resistance [74]. Using single-cell analysis on ileal resection of CD, Martin et al. found that activated fibroblasts expressing PDPN are part of a GIMATS module which is constituted of inflammatory macrophages, activated dendritic cells, activated T cells, and activated fibroblasts, whose presence is associated with resistance to anti-TNFα treatment in a pediatric cohort (RISK cohort), corroborating previous results [75]. Moreover, Smillie et al. showed the upregulation of genes encoding for IL-11, IL-25, IL-13RA2, and fibroblast activation protein (FAP) in activated fibroblasts from UC patients [76]. Another example of activated fibroblasts’ modulation of the immune response is that fibroblasts isolated from CD patients produce a high level of MMP10 leading to the decrease of the membrane-bound form of the immune-negative regulator PDL1 at their surface and, consequently, an increase of the immune activity [77]. Strikingly, this increased expression of MMP10 is not observed in UC patients.

Fibroblasts isolated from UC and CD patients display reduced migration capacity, accompanied by a decrease in focal adhesion kinase phosphorylation [78]. The enhancement of the cell motility is a proposed effect of anti-TNFα treatments such as infliximab or adalimumab antibodies, accompanied with an increase in tissue inhibitor of metalloproteinase (TIMP) -1, as demonstrated in colonic CD fibroblasts [79]. IBD fibroblasts are more proliferative than the ones isolated from healthy tissues, with an increased production of collagen type 1. However, no difference is observed between fibroblast populations from UC or CD samples [80]. During IBD, activated fibroblasts overproduce Lysyl oxidase (LOX) gene family, LOX and LOXL1 [29], therefore, enhancing the oxidative stress-mediated inflammation as a result of hydrogen peroxide generation [81]. Indeed, inhibition of LOX by β-aminopropionitrile treatment in colitis murine model results in a drastic improvement of the animal’s condition. Moreover, in vitro, LOX overactivity induces a contraction of the tissue and decreases MMP3 production in stenotic fibroblasts isolated from CD patients [82]. ECM remodeling by fibroblasts results in an increased elasticity in inflammatory colonic tissues, which in turn enhances proliferation, αSMA expression, and pro-inflammatory gene expression [83], therefore, participating in the auto-propagation of the intestinal fibrosis. The inflammation perpetuation can also be linked to the enhanced secretion of pro-angiogenic factors by the fibroblasts, therefore stimulating neo-angiogenesis, allowing the inflammatory cells to access the inflammation site [84]. Increased angiogenesis can actually also be mediated through ECM components such as laminin, fibronectin, or collagen type IV-derived matrikines, which have pro- or anti-angiogenic properties and whose expression regulation is altered in IBD and CRC [85].

## 4. From Sustained Inflammation to Tumorigenesis

### 4.1. How It All Starts

For years, CRC development dogma implicated the emergence of successive genetic alterations leading to tumorigenesis, described as the “adenoma–carcinoma sequence”. The well-characterized mutagenic sequence in CRC begins with an inactivating mutation in the *APC* gene, a key downregulator of the Wnt/β-catenin proliferative pathway, and continues with the acquisition of subsequent mutations of oncogenes, such as *KRAS, PI3KCA, SMAD4, and TP53*, forming a malignant tumor [86]. A number of studies proposed ISC as cells of origin for CRC since activating mutations of the Wnt/β-catenin pathway in LGR5^+^, Bm1^+^, or Prom1^+^ cells lead to adenoma development in mice [5,6,7]. However, such mutations in differentiated cells form stalling microadenomas, suggesting that alternative factors are necessary for tumor initiation. Vermeulen et al. proposed a model based on a competition between mutants and wild-type (WT) ISC for the occupation of the crypt [87]. In this model, mutant cells have a prominent, but not total probability, to conquer the crypt depending on multiple parameters including bottom crypt position. Interestingly, they calculated that a dominant negative mutated TP53 clone has a competitive advantage versus WT only in the context of colitis. Here, mutated ISC selection depends on physiological ISC niche spatial occupancy and dominance. Moreover, as previously said, several clinical investigations demonstrated an increased risk for CRC apparition in patients subject to IBD [88,89,90,91], although some studie contested this affirmation [92]. As a matter of interest, CRC risk appears to be dependent on the type, the duration, and the severity of the pathology. It strongly suggests that the microenvironment of the crypt is involved. The colorectal crypt environment during chronic inflammation is highly disturbed. Persistent activated fibroblasts and immune cells are concentrated near the epithelium, where they intensively secrete growth factors, pro-survival molecules, pro-inflammatory cytokines, and ECM components to stimulate tissue repair and fibrosis. In this way, they also create the breeding ground for tumorigenesis. Tumor initiation occurring in a physiological context (stochasticity, i.e., sporadic CRC) or in an inflammatory or regenerative context (i.e., colitis-associated cancer (CAC)) is wildly different and could explain differences between these two types of CRC development. However, in this manuscript we will not develop the occurrence of sporadic colorectal cancer and the role of CAF in this tumorigenesis since it has been recently very nicely reviewed [11].

Studies on animal models revealed that chronic inflammation sensitizes CRC development after cancerogenic treatment [93]. Moreover, mice lacking the pro-inflammatory interleukin IL-6 display a decreased occurrence in CAC tumorigenesis [94]. By contrast, mice lacking the anti-inflammatory cytokine IL-10 spontaneously developed colorectal tumors after colitis [95]. Inactivation of IL-10 is also observed in children with IBD [68], and anti-inflammatory treatments have been proposed as preventives therapies for CRC in IBD patients [96,97]. Considering that inflammation involves many players, it is difficult to decipher whether and how each are implicated in CAC. As strong producers of interleukins and inflammatory cytokines, activated fibroblasts have an important responsibility in the colorectal tumorigenesis. For example, in the IL-10^-/-^ mice, COX2, which is overexpressed by myofibroblasts, is strongly increased and associated with the ulcerated regions of colorectal tumors [95]. Two recent elegant studies made by Mechta-Grigoriou’s team on breast and ovarian cancers provided an overview of how fibroblast populations can be the main actors in tumor development and can drive the resistance to cancer therapies [98,99]. Using single-cell analysis, they were able to divide fibroblasts into clusters, some of which were related to immunosuppression, cancer aggressiveness, and resistance to immunotherapy. Those fibroblasts attract CD4^+^CD25^+^ T cells through the secretion of CXCL12, retain them on contact via ligand for OX40 (OX40L- OX40 also known as Tumor necrosis factor receptor superfamily, member 4 (TNFRSF4)), Programmed cell death 1 ligand 2 (PD-L2), and Junctional adhesion molecule B (coded by the gene *JAM2)*, and then promote survival and differentiation into Treg FOXP3^+^, thus inhibiting T effector proliferation. The ratio of each cluster is patient dependent, therefore, considering that such types of subpopulations exist in all organs, it may explain the discrepancy observed in clinical analysis regarding the risk of CAC in IBD patients.

In parallel, activated fibroblasts contribute to the stiffening of the tissue by remodeling the interstitial matrix. Increased elasticity has been measured not only in inflamed colon samples [83] but also in CRC samples, in which elasticity correlates with the CRC stage [100]. Interestingly, αSMA staining also positively correlates with CRC development [100], indicating a continuous increase in the activated-fibroblast population. Matrix stiffness has a prominent role in the control of the physiological behavior of stem cells [101]. Indeed, culturing mesenchymal stem cells on soft or stiff substrates, it has been shown that stiffness by itself can drive the differentiation toward peculiar lineages and promotes cells proliferation [102,103,104,105]. The implication of stiffness in the epithelial-to-mesenchymal transition, known to enhance cancer progression and invasion, has been demonstrated in vitro [97,106,107]. However, authors used cancer cells lines, thus, introducing a bias in the analysis and not deciphering the real implication of stiffness in the switch from a physiological to a cancerous phenotype. Nevertheless, one can suppose that increased stiffness during chronic inflammation may results in an alteration of ISC homeostasis, thus, leading to dysregulation in proliferation rate and conversion toward a tumorigenic phenotype.

Moreover, stroma alterations, occurring for instance during inflammation, could also indirectly affect the epithelial cell in close proximity to these activated fibroblasts. As a matter of fact, one of the strongest supports in regard to the role of microenvironment alteration in colorectal tumor initiation is the long-standing observation that chronic inflammation, such as that found in Crohn’s disease (CD) or ulcerative colitis (UC), is a significant risk factor for CRC [108]. Patients suffering from inflammatory bowel diseases (IBD, including CD and UC) have a high risk of developing colitis-associated CRC (CAC) and have high mortality from these diseases [108,109]. More importantly, in the majority of patients who did not show signs of IBD pathogenesis prior to CRC onset, tumor-associated inflammation is evident in clinical samples and has been shown to drive cancer development in the gut, suggesting a fundamental role for inflammation in both CAC and sporadic CRC development [108]. Quante et al. report that an inflammatory microenvironment could affect tumor initiation in intestinal stem cells [109]. Crypt stem cells have been identified as the cells of origin of intestinal cancer in colitis patients [110]. Moreover, in the field of tumor-initiating cells, recent data have focused attention on precancerous stem cells characterized by impaired self-renewal, multi-potency of differentiation, and requirement of a pro-tumorigenic niche for their maintenance [111].

### 4.2. Cancer-Associated Fibroblasts: Colorectal Cancer’s Band Leader

The incidence of CAFs in tumor progression is better understood; however, it becomes more complex as the knowledge progresses [112,113,114]. Indeed, CAFs promote cancer development through tumor environment remodeling, cancer stemness enhancing, cancer cell feeding, and immune response suppression, but evidence shows that CAFs can also turn into tumor inhibitors. This suggests, and it was confirmed by recent studies, that CAF populations are heterogeneous [98,99]. A recent study demonstrated the importance of two CAF subpopulations involved in BMP signaling regulation that ensure the balance between stemness enrichment and tumor cell differentiation in the mouse and human CRC organoid model [115]. A first population, associated with a poor prognosis in human CRC, is characterized by the increased secretion of GREM1, the BMP antagonist known to play a predominant role in the physiological ISC niche. By analogy with fibroblasts’ role in the physiological regulation of the stem cell population, in CRC, the CAF population secreting GREM1 is associated with tumor stemness. Conversely, the CAF population described to promote differentiation and to block stemness is characterized by enrichment in immunoglobulin superfamily containing leucine-rich repeat (ISLR) protein causing BMP pathway stimulation and is, thus, associated with a good prognosis in CRC. The role of CAF in CRC progression and aggressiveness has become prominent thanks to transcriptomic analysis demonstrating that poor prognosis and resistance to radiotherapy correlate with a gene signature linked to stromal cells rather than epithelial tumor cells [116,117].

According to the robust classification system for CRCs established by Guinney et al., four distinct classes can be described with particular features based on metabolism pathways, chromosomal instability, mutational status, or stromal cells invasion [118]. One of them (called consensus molecular subtypes 4 or CMS4), representing 23% of CRCs, is characterized by a strong enrichment in stromal cells associated with TGFβ signaling activation. Interestingly, transcriptomic signature associated with CMS4 shows the overexpression of genes associated with ECM remodeling, EMT, and TGFβ activation. All of these features are strangely associated with processes involving fibroblast populations in colonic pathophysiology (fibroblasts role, activated fibroblasts upcoming via EMT, or resident activation through TGFβ). We can, therefore, assume that CMS4 represents a class of CRCs where fibroblasts are the main drivers of tumorigenesis. Unlike CMS1, which is mainly enriched with activated immune cells (Th1 and PD1 activation, NK cells infiltration, and decreased T regulators lymphocytes signature), it is only the complement system activation that is increased in CMS4. These results correlate with other studies on CAF immunosuppressive capacities. Through their immune-suppressive capacities, CAFs generate an immune free environment surrounding the tumor. For example, in mouse models of immune-excluded breast and colorectal cancers, a co-treatment with anti-TGFβ and anti-PDL1, immune-suppressor expressed by CAFs, results in a decrease in matrix-remodeling factors, an increase in T cell reaching the tumor site, and tumor regression [119,120]. All of these data may suggest that depending on the CRC microenvironment and subtype (patient history, chronic inflammation, chromosomal instability, mutations, etc.), CAFs are more or less abundant and may or may not promote the tumor cells’ escape from the immune system’s surveillance. As in physiology or in wound healing, increasing evidence seems to say that different CAF populations can co-exist and exert either a pro- or anti-inflammatory action on the tumor microenvironment.

CAFs are also implicated in feeding tumor cells as they are able to perform autophagy and mitophagy to release energy-rich metabolites later absorbed by cancer cells that undergo oxidative mitochondrial metabolism [121,122]. In parallel, discrepancies appear suggesting a protective role of CAFs against tumor development. In a pancreatic ductal adenocarcinoma (PDAC) mouse model, the inhibition of Sonic Hedgehog (Shh) in epithelial cells surprisingly results in tumor progression, in part reversed by a vascular endothelial growth factor receptor (VEGFR) blocking antibody, indicating that epithelial/stromal cells’ communication through Shh pathways promotes high stroma content in PDAC and acts to restrain the tumor at least in part through the inhibition of angiogenesis [123]. In the same model, αSMA^+^ stromal cell deletion accelerates PDAC progression, while enhancing EMT and cancer stem cells [124]. Moreover, Shh signaling is decreased in CRC, and its restoration reduced tumor progression [125]. As for IBD, angiogenesis is known to participate in CAC progression. For instance, in a model of CAC-derived from DSS-treated mice, transcriptional analysis revealed that *tenascin C* gene expression is markedly increased in the stroma surrounding dysplastic lesions. The inhibition of tenascin C/αvβ3 integrin interaction suppresses CAC via the inhibition of angiogenesis [126].

## 5. Conclusions

Understanding the dialog between intestinal epithelial cells and their surrounding environment is critical to understand the apparition and development of pathologies such as cancer. Fibroblasts, as stromal major cell population, are key partners for ISC homeostasis. More and more clues have allowed us to grasp the entirety of their roles, i.e., the physiological, the “under control-pathological”, and the “uncontrolled-pathological” ones. (i) They are the main producers of proliferative and anti-differentiation factors at the bottom of the crypt and pro-differentiation and anti-proliferative molecules at the top, therefore actively contributing to intestinal physiological integrity. (ii) After injury, their role gains importance as the main leader of epithelium healing and regeneration through pro- and anti-inflammatory molecule and growth factor secretion. (iii)They participate in cancer progression, for example, by increasing tissue stiffness and promoting tumor cell migration and proliferation. However, the puzzle is far more complicated, especially when considering CRC initiation. Indeed, CRC initiation is characterized either by an epithelium-to-stroma (e.g., CRC) or, on the contrary, by a stroma-to-epithelium (e.g., CAC) pathological scheme, in which it is important to discern the different fibroblasts’ contributions.

Sporadic CRC is one of the most understood cancer in terms of initiation and development. The genetic sequence dysregulation begins in the intestinal epithelial stem cells by an inactivating mutation of *APC* allowing the formation of a small adenoma, follows by activation of oncogenes such as *KRAS*, initiating cancer cell expansion. The expansion process continues by mutations in genes such as *PI3K, TP53*, or *SMAD4.* The tumor cells induce in return biochemical and biophysical pressure on their surrounding ECM, leading to important remodeling, such as matrix and metalloproteinase protein secretion, for which the activation of fibroblasts is necessary. The role of activated fibroblasts in cancer progression is being understood more and more and led to the discovery of fibroblast subpopulations, with either a protective or an aggressive role in the pathology. Therefore, in sporadic CRC, fibroblast activation occurs at least in part in response to the ISC tumorigenic switch.

Interestingly, in colorectal cancer appearing after chronic episodes of inflammation and repair, e.g., CAC, the cause/effect process of cancer initiation appears to be somehow inverted. Indeed, the sustained remodeling of the stroma, including fibroblasts and immune system activation, production of ECM fibers increasing tissue stiffness, and the persistent production of growth factors and proliferative and anti-apoptotic molecules, induces a constant stress on the colonic epithelial cells. This stress abolishes ISC homeostasis and results in the selection of ISC with genetic alterations not fully understood but different than those observed for sporadic CRC, such as epigenetic modifications or mutations in key regulatory genes as *TP53*, which in turn lead to the development of a tumor.

This concept seems increasingly realistic, and recent findings highlighting the presence of fibroblast subpopulations not only in colorectal cancer but also at physiological state, paving the way toward a better understanding of their part in the maintenance of the intestinal integrity or the disease progression. However, much remains to be elucidated to fully understand the contribution of fibroblasts and activated fibroblasts in the different steps of the pathology, as well as to establish a signature allowing discriminating fibroblast subpopulations, and finally, to decipher their participation in the balance between the regeneration and cancer initiation processes.

## Figures and Tables

**Figure 1 cancers-13-01749-f001:**
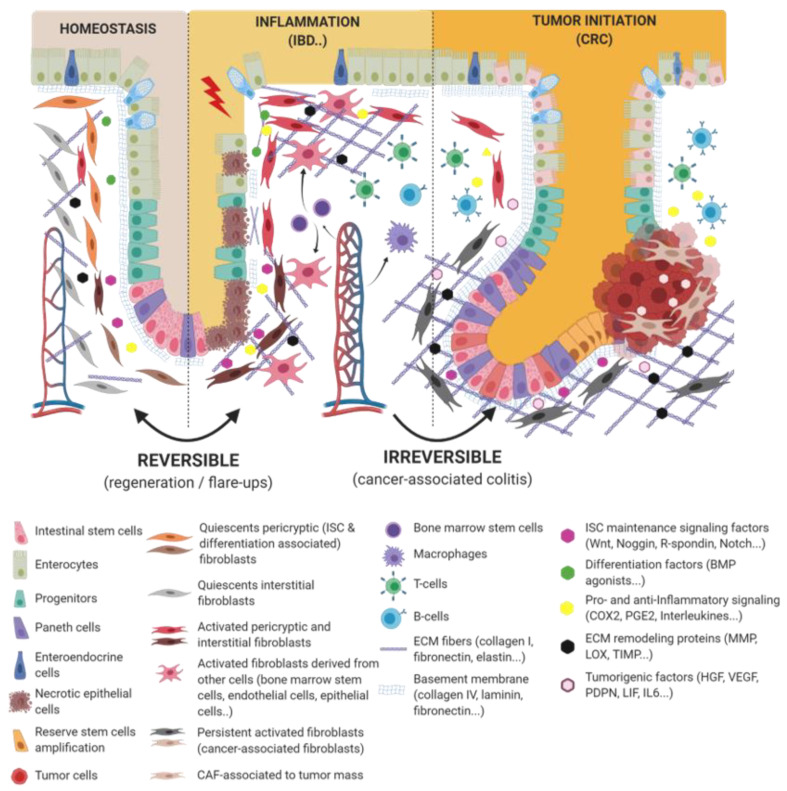
Epithelio-fibroblastic relationship in the physiological and injury context in intestinal tissue. Under the physiological context, normal fibroblasts form a very heterogeneous cell population that could be divide into two families: interstitial fibroblasts (extracellular matrix (ECM) production and regulation) and pericryptic fibroblasts (soluble factors secretion and ECM basement membrane production). Depending on pericryptic fibroblasts’ localization on the intestinal crypt compartment, they secrete (i) proliferating factors for intestinal stem cell (ISC) maintenance (wingless-related integration site (Wnt ligands, R-spondin, EGF, bone morphogenic protein (BMP) antagonists) and constitute an ISC niche or (ii) differentiating factors (BMP family). In the case of acute or chronic tissue injury, not only epithelial cells but also fibroblastic populations undergo a strong remodeling, resulting in ISC loss and/or inability to regenerate epithelium. Depending on the injury severity, resident fibroblasts can be lost or become transiently or definitively activated. Other cell types can differentiate and reach an activated-fibroblast phenotype, thus promoting epithelial regeneration, survival, and immune dialog. Activated fibroblasts can improve epithelial regeneration on a first intention (by increasing some ISC niche factors’ production) and then switch their phenotype to impaired ISC and epithelium renewal.

**Figure 2 cancers-13-01749-f002:**
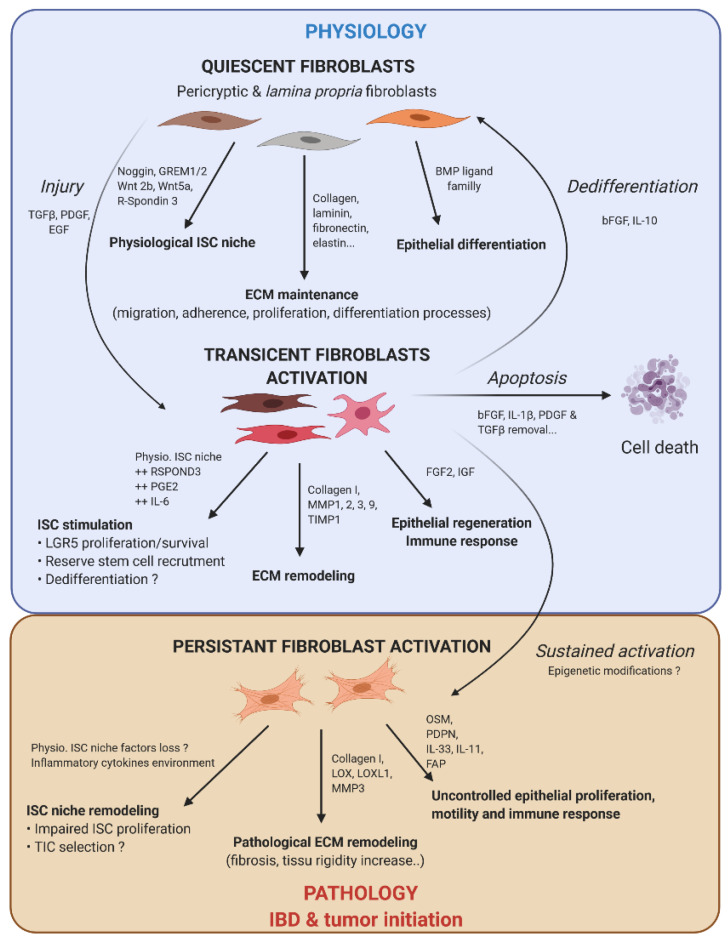
Fibroblast activation dynamic switch depending on the physiopathological context. Normal fibroblasts have a pivotal role under the physiological context to produce ECM protein, control ISC compartment proliferation/differentiation, and maintain intestinal tissue homeostasis. Fibroblasts are very plastic cells able to respond to their environment by becoming activated in case of injury to support ISC proliferation and renewal. To compensate eventual ISC loss, the intestinal epithelium can call on reserve stem cells, which can be recruited by stromal stimulation, in particular prostaglandin E2 (PGE2), or a dedifferentiation of secretory or fully differentiated progenitors, to join the compartment strain and thereby regenerate the epithelium. After epithelium repair, the activated fibroblasts can either become quiescent again, or regain their physiological function (by dedifferentiation), or die by apoptosis. However, if the injury persists and becomes chronic, the activated fibroblasts are locked into this state (possibly due to changes in the epigenetic status). This is where activated fibroblasts can play an important role in the emergence or initiation of new pathologies such as cancer.

**Figure 3 cancers-13-01749-f003:**
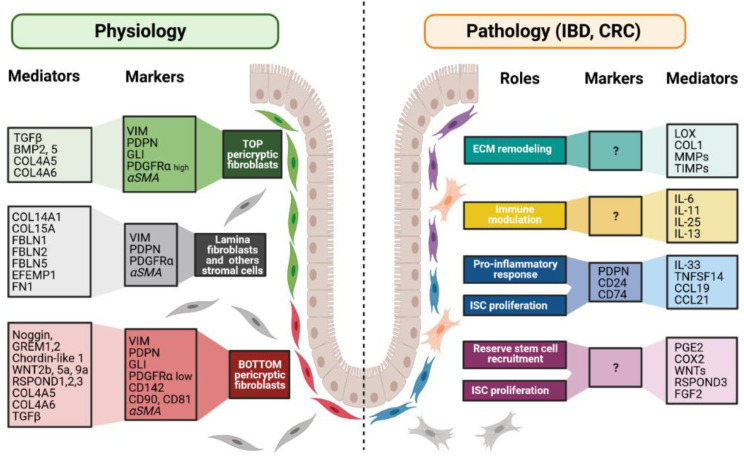
Fibroblast markers and mediators under physiological and pathological contexts. Fibroblasts can be divided into different populations depending on their locations, functions along the crypt axis, and pathophysiological status. More and more studies try to present a panel of markers that can be used to decipher those populations. Although none of them are actually specific of a subpopulation, in combination, they allow the discrimination of the different subpopulations, namely top pericryptic-, bottom pericryptic- and lamina propria-associated fibroblasts. However, discrepancies exist for example concerning alpha smooth muscle actin (αSMA), which can be found, or not, on the different populations. Under the pathological context, the discrimination of the diverse fibroblasts populations is increasingly complex, with no markers being currently known to clearly identify the ones implicated in the various pathological processes. Markers and mediators listed in this figure are used as examples and are not exhaustive.

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
