# Peer review of "Colon Fibroblasts and Inflammation: Sparring Partners in Colorectal Cancer Initiation?"

_cancers, 2021, doi:10.3390/cancers13081749_

Round 1

Reviewer 1 Report

In this manuscript Onfroy-Roy and colleagues review the current status of knowledge regarding the role of fibroblasts and inflammation in the initiation of colorectal cancer. The authors first explain the physiological role of fibroblasts in the crypt and  describe the underlying mechanism that drive their activation. They further review how persistent fibroblast activation can lead to chronic inflammation which in turn can lead to tumorigenesis.

Overall the review has been structured well, with a good flow and the chronological manner of structuring makes it easy to go back and forth. The joyful titles make it easy to follow and cover the content well. However I do have an important suggestion for the title of the review: I believe you mean to say ‘Sparring partners’ and not ‘Sparing partners’ as this makes a major difference in the meaning of the title.

Although the review is well-structured it also contains rather lengthy sentences which does not benefit the readability of the review. Some sentences need shortening and more figures would certainly help (see further remarks).

Specific points:
Overall, the introduction is quite long and should be shortened. The purpose of the review is lost after reading such a long introduction. Please makes this pointier and more concise

Sentences 34 and 35 contain a conclusion that cannot be substantiated and should be removed. ‘…,meaning that currently available treatments are unable to eliminate all the cancer cells’. Lot’s of patients are cured very effectively by surgery. Others are cured by (neo) adjuvant chemo or radiation, so this conclusion is in essence wrong as in a large part of the patients all tumor cells are effectively eliminated.  

Sentences 100-101: ‘Activated fibroblasts are… cells in vivo’. You cannot conclude from the fact that activated fibroblasts are involved in proliferation and dedifferentiation that they thus convey resistance to therapy. This should be weakened and importantly references to primary papers that support this idea should be included.

In section 2 a lot of genes are discussed and due to the complexity and diversity of expressed genes the data is cluttered and unclear. Of course different papers suggest different subtypes of fibroblasts with a heterogeneous expression, hence description will invariably be difficult to follow, but figures and/or tables supporting the text would be very helpful for the review. Furthermore, a sentence or two concluding this very long paragraph in a way that the authors try to compile all the insight into digestible conclusions would be very helpful to lead the reader by the hand.

The transition towards inflammation is rather abrupt. A smoother transition into this section with an introduction on the origins of inflammation and/or injuries would be needed. Now it is just mentioned that inflammation and injuries exist, but how do these come about? Once again this is a large paragraph with a lot of interesting data, but it could be organized better.

  1. From sustained inflammation to tumorigenesis
    There are distinctive subtypes in colorectal cancer and they are ignored completely in this review. Especially in this paragraph they should be mentioned regarding inflammation and stromal involvement. Also a transition of this chapter to the next about CAFs is lacking. How do fibroblasts turn into CAFs.? As the review is about tumor initiation it would be relevant to discuss CAF formation.

    In part 2.2 you mention in that through their immune suppressive capacities, CAFs generate an immune free environment surrounding the tumor. However, in the mesenchymal CMS4 cancers (Guinney, Nature Medicine 2015) there is a high stromal infiltration as well as a high immune infiltration. The authors should use their data and review to explain that this is an immune suppressive environment.

Finally, the discussion is solid but a couple of summarizing sentences would be helpful.

Minor points

Overall grammar, spelling and readability:
There are quite some typos and spelling errors in this review which make the review harder to read. I will point out a few but I suggest to re-read the paper to get all of them out:
sentence 143: favors = favor
sentence 149: distinct = distinguish
sentence 343: Wound = Wounds
sentence 451: contributes = contribute
sentence 468: CAF = CAFs
sentence 513: gain = gains
sentence 515: participates = participate
sentence 519: in a contrary = on the contrary

Author Response

Response to Reviewer 1 (R1):

Thank you very much for your comments and suggestions on our manuscript. We took all of them into consideration and have now made the suitable revisions and modifications of the article in the light of the your comments as outlined on the attached file to answer the different points raised. We trust that this version of the manuscript will be found suitable for publication in Cancers.

Title:

 1: As correctly mentioned by Reviewer 1 (R1), we changed the title from ‘Colon fibroblasts and inflammation: Sparing partners in colorectal cancer initiation?’ to ‘Colon fibroblasts and inflammation: Sparring partners in colorectal cancer initiation?’

Specific points

1: Overall, the introduction is quite long and should be shortened. The purpose of the review is lost after reading such a long introduction. Please makes this pointier and more concise

We agree with this comment. The introduction has been shortened

2: Sentences 34 and 35 contain a conclusion that cannot be substantiated and should be removed. ‘…,meaning that currently available treatments are unable to eliminate all the cancer cells’. Lot’s of patients are cured very effectively by surgery. Others are cured by (neo) adjuvant chemo or radiation, so this conclusion is in essence wrong as in a large part of the patients all tumor cells are effectively eliminated.  

We agree on this comment and deleted this end of sentence mentioning that current treatments are not able to eliminate all the cancer cells.

.  

3: Sentences 100-101 (now 104-106): ‘Activated fibroblasts are… cells in vivo’. You cannot conclude from the fact that activated fibroblasts are involved in proliferation and dedifferentiation that they thus convey resistance to therapy. This should be weakened and importantly references to primary papers that support this idea should be included.

            We rephrased here, and regarding the fibroblast participation to the resistance, we wrote ‘can participate to tumor resistance for example by increasing the ECM stiffness around the tumor, or by upregulating the epithelial stemness capacities and favoring the epithelial-mesenchymal transition in vivo’. References have been added.

4: In section 2 a lot of genes are discussed and due to the complexity and diversity of expressed genes the data is cluttered and unclear. Of course different papers suggest different subtypes of fibroblasts with a heterogeneous expression, hence description will invariably be difficult to follow, but figures and/or tables supporting the text would be very helpful for the review.

            We created the figure 3, presenting fibroblast markers and mediators under physiological and pathological contexts.

5: The transition towards inflammation is rather abrupt. A smoother transition into this section with an introduction on the origins of inflammation and/or injuries would be needed. Now it is just mentioned that inflammation and injuries exist, but how do these come about?

            We agree on this comment and add the notion that chronic inflammation such as in inflammatory bowel disease can lead to cancer. We add this little paragraph:’ The inflammatory bowel diseases, which two main subtypes are Crohn’s disease (CD) and ulcerative colitis (UC), are chronic relapsing inflammatory disorders of the gas-trointestinal tract. IBD afflicts several millions of persons worldwide. Their incidence is high among developed countries and increases steadily. These diseases are characterized by an excessive immune response to gut microbiota dysbiosis, but also by an important impairment of the epithelial renewal process, leading to an abnormal mucosal re-pair/healing. Indeed, a major complication of UC is its evolution into cancer.’

  1. From sustained inflammation to tumorigenesis: There are distinctive subtypes in colorectal cancer and they are ignored completely in this review. Especially in this paragraph they should be mentioned regarding inflammation and stromal involvement.

                We replaced in this section the paragraph ‘Moreover, stroma alterations, occurring for instance during inflammation, could also indirectly affect the epithelial cell in close proximity to these activated fibroblasts. As a matter of fact, one of the strongest supports in regard to the role of microenvironment al-teration in colorectal tumor initiation is the long-standing observation that chronic in-flammation, as found in Crohn’s disease (CD) or ulcerative colitis (UC), is a significant risk factor for CRC [108]. Patients suffering from inflammatory bowel diseases (IBD, in-cluding CD and UC) have a high risk of developing colitis-associated CRC (CAC) and have high mortality from these diseases [108,109]. More importantly, in the majority of patients who did not show signs of IBD pathogenesis prior to CRC onset, tumor-associated inflammation is evident in clinical samples and has been shown to drive cancer development in the gut, suggesting a fundamental role for inflammation in both CAC and sporadic CRC development [108]. Quante et al. report that an inflammatory microenvironment could affect tumor initiation in intestinal stem cells [109]. Crypt stem cells have been identified as the cells-of-origin of intestinal cancer in colitis patients. Moreover, in the field of tumor-initiating cells, recent data have focused attention on precancerous stem cells characterized by impaired self-renewal, multi-potency of differentiation, and requirement of a pro-tumorigenic niche for their maintenance [111].’

6: In part 2.2 you mention in that through their immune suppressive capacities, CAFs generate an immune free environment surrounding the tumor. However, in the mesenchymal CMS4 cancers (Guinney, Nature Medicine 2015) there is a high stromal infiltration as well as a high immune infiltration. The authors should use their data and review to explain that this is an immune suppressive environment.

                We agree and developed this section by adding ‘According to the robust classification system for CRCs established by Guinney et al., four distinct classes can be described with particular features based on metabolism pathways, chromosomal instability, mutational status or stromal cells invasion [118]. One of them (called consensus molecular subtypes 4 or CMS4), representing 23% of CRCs, is characterized by a strong enrichment in stromal cells associated with TGFβ signaling activation. Interestingly, transcriptomic signature associated with CMS4 shows genes overexpression associated with ECM remodeling, EMT and TGFβ activation. All of these features are strangely associated with processes involving fibroblast populations in colonic pathophysiology (fibroblasts role, activated fibroblasts upcoming via EMT or resident activation through TGFβ). We can therefore assume that CMS4 represents a class of CRCs where fibroblasts are the main drivers of tumorigenesis. Unlike CMS1, which is mainly enriched with activated immune cells (Th1 & PD1 activation, NK cells infiltration and decreased T regulators lymphocytes signature), it is only the complement system activation that is increased in CMS4. These results correlate with other studies on CAF immunosuppressive capacities’

Minor points

7 : Typos and spelling issues have been corrected and revised.

Reviewer 2 Report

Dear authors, 

The paper entitled "Colon fibroblasts and inflammation: Sparing partners in colorectal cancer initiation?" is a good review, interesting, well structured, and with great section names. 

However, I miss some comments about the role of oxidative stress (present in an  chronic inflammation scenario) in colon carcinogenesis (i.e promoting DNA mutations).

By other hand, correct the following:

  1. In the Abstract, it's said that CRC is the second cause of cancer mortality, and in the Introduction it's indicated that CRC is the third most common cause of cancer-related death. 
  2. In Figure 1, there are purple-colored cells in the epithelium named as Paneth cells, but there are no Paneth cells in the crypts of large intestine, only in small intestine. So, please, remove Paneth cells from this figure. 

Author Response

Thank you very much on your comments on our manuscript. We have now made suitable revisions and modifications of the article in the light of the reviewer’s comments. English grammar and typos have been revised. Modifications are highlighted in the manuscript. We trust that this version of the manuscript will be found suitable for publication in Cancers.